# Investigation on the Photodegradation Stability of Acrylic Acid-Grafted Poly(butylene carbonate-co-terephthalate)/Organically Modified Layered Zinc Phenylphosphonate Composites

**DOI:** 10.3390/polym15051276

**Published:** 2023-03-02

**Authors:** Yi-Fang Lee, Tzong-Ming Wu

**Affiliations:** Department of Materials Science and Engineering, National Chung Hsing University, 250 Kuo Kuang Road, Taichung 40227, Taiwan

**Keywords:** photodegradation, biodegradable polymer, composites, UV protection, photodegradation stabilizer

## Abstract

The application efficiency of biodegradable polymers used in a natural environment requires improved resistance to ultraviolet (UV) photodegradation. In this report, 1,6-hexanediamine modified layered zinc phenylphosphonate (m-PPZn), utilized as a UV protection additive for acrylic acid-grafted poly(butylene carbonate-co-terephthalate) (g-PBCT), was successfully fabricated and compared to the solution mixing process. Experimental data of both wide-angle X-ray diffraction and transmission electron microscopy reveal that the g-PBCT polymer matrix was intercalated into the interlayer spacing of m-PPZn, which was approximately delaminated in the composite materials. The evolution of photodegradation behavior for g-PBCT/m-PPZn composites was identified using Fourier transform infrared spectroscopy and gel permeation chromatography after being artificially irradiated by a light source. The change of carboxyl group produced via photodegradation was used to show the enhanced UV protection ability of m-PPZn in the composite materials. All results indicate that the carbonyl index of the g-PBCT/m-PPZn composite materials after photodegradation for 4 weeks was extensively lower than that of the pure g-PBCT polymer matrix. These findings were also supported by the decrease in the molecular weight of g-PBCT after photodegradation for 4 weeks, from 20.76% to 8.21%, with the loading of 5 wt% m-PPZn content. Both observations were probably owing to the better UV reflection ability of m-PPZn. This investigation shows, through typical methodology, a significant advantage of fabricating the photodegradation stabilizer to enhance the UV photodegradation behavior of the biodegradable polymer using an m-PPZn compared to other UV stabilizer particles or additives.

## 1. Introduction

The traditional synthetic polymers, such as polyamide (PA), poly(ethylene terephthalate) (PET), polypropylene (PP), and polystyrene (PS), fabricated from petrochemical products have been shown to have environmental influences owing to their poor degradability and recovery/reproduction rates. In contrast, there has recently been a renewal of interest in biodegradable polymers, such as poly(lactic acid) (PLA), poly(butylene adipate-co-terephthalate) (PBAT), poly(butylene carbonate-co-terephthalate) (PBCT), poly(1,4-butanediol succinate) (PBS), and poly(butylene succinate-co-terephthalate) (PBST), due to their environmentally friendly characteristics and their use to substitute conventional fossil-based polymer materials in applications such as agriculture, biomedical materials, energy harvesting composite materials, packaging, and textiles [1,2,3,4]. Among these biodegradable polymers, the aliphatic components of aliphatic-aromatic PBAT, PBST, and PBCT have shown capable biodegradability, and the aromatic components in their polymer backbones posse outstanding mechanical properties. Compared with aliphatic-aromatic PBAT and PBST, PBCT contains more promising comprehensive properties and competitive cost with a low crystallization rate and melting temperature, which makes it a possible contender for applications in the biomedical and package fields due to the absence of acidic compounds during its in vivo degradation and photodegradation [5]. The most effective abiotic degradation happening in the environment is photodegradation. Various measurements are employed to examine the influences of the polymer’s disclosure to sunlight. The easiest method to realize, least expensive, and closest to real-world circumstances is outdoor exposure [6]. In order to protect against adverse environmental effects, most synthetic polymers require stabilization. Therefore, it is crucial to discover a method to avoid or decrease the damage produced by environmental factors; for instance, heat, light, or oxygen [7].

The mechanical properties and melt processability of PBCT have been significantly enhanced using branching architecture in the polymer backbone, which could enhance its business opportunities for compostable packaging applications [8]. Graft polymerization is one of the simplest approaches to prepare a material with functional groups to the surface between the substrate and the grafted layer [9]. Among these reactions, acrylic acid containing a carboxyl functional group is commonly used in such grafting processes. In order to further inhibit the photodegradation properties of PBCT copolymer, the incorporation of the outstanding UV absorption properties of an inorganic material between 250 and 400 nm, such as layered zinc phenylphosphonates (PPZn), with a two-dimensional lamellar structure, acted as the protecting material into the polymer matrix and enhanced its photo barrier properties compared to the zero-dimensional UV stabilizer particles or additives [10,11]. Since PPZn contains a coordinated water molecule in the interlayered spacing, the loss of this water molecule provides possible intercalation or catalytic reactions with an open coordination site on the zinc atom [12,13]. Thus, organo-modifiers are necessary to enlarge the interlayered spacing of PPZn and to generate better compatibility between the polymer matrix and PPZn [14]. According to this study, the incorporation of PPZn can significantly enhance the crystallization and enzymatic degradation rate of the fabricated nanocomposites.

Acrylic acid-grafted PBCT (g-PBCT)/organically modified layered zinc phenylphosphonate (m-PPZn) nanocomposites with covalent bonding between the polymer and reinforcing materials were successfully synthesized. In this study, we focus on the UV protection effect of m-PPZn on g-PBCT. To stimulate the photodegradation of the g-PBCT/m-PPZn samples, an artificial light source was provided. From our detailed research, no investigation on the photodegradation of the g-PBCT/m-PPZn nanocomposites has been reported, thus, this is novel research. The carbonyl index, number-average molecular weight, and morphology of prepared samples were used to estimate the UV protection of m-PPZn in the g-PBCT polymer matrix.

## 2. Experimental Materials

Phenylphosphonic acid (C_6_H_5_P(O)(OH)_2_, >98%), sodium hydroxide (NaOH, >98%), zinc nitrate hexahydrate (Zn(NO_3_)_2_·6H_2_O, >99%), 1-ethyl-3-(3-dimethylaminopropyl)carbodiimide (C_8_H_17_N_3_·HCl, EDC, >98%), and 1,6-hexanediamine (H_2_N(CH_2_)_6_NH_2_, >97%) were obtained from Sigma-Aldrich (St. Louis, MO, USA). Acrylic acid (C_3_H_4_O_2_, AA, >99%) was acquired from Tedia Company, Inc. (Fairfield, MO, USA). 2,2′-Azobis-isobutyronitrile (C_8_H_12_N4, AIBN, >98%), chloroform (CHCL_3_, >99%) was purchased from Showa Chemical Industry Co., Ltd. (Minato-Ku, Tokyo, Japan). 1,4-butanediol (C_4_H_10_O_2_, BD, >99%), dimethylene carbonate (C_3_H_6_O_3_, DMC, >98%), and dimethylene terephthalate (C_10_H_10_O_4_, DMT, >99%) were obtained from Alfa Aesar Chemical Company (Ward Hill, MA, USA). All chemicals were utilized as received.

### 2.1. Fabrication of g-PBCT/m-PPZn Composites

PBCT synthesized using the transesterification and polycondensation process has been previously reported [5,15,16,17,18,19]. In brief, proper amounts of BD, DMC, DMT, and sodium hydroxide used as a catalyst were mixed and mechanically stirred. The mixtures were heated to 120 °C under nitrogen gas for 1 h, then heated to 190 °C for 1 h, and finally heated to 220 °C for 4 h under vacuum. The fabricated PBCT was dissolved in dichloromethane, and then precipitated from cold methanol for purification. The above experiments needed to be repeated three times to complete the purification process. To allow the grafting reaction to occur, the obtained PBCT was dissolved in a mixture of AIBN and AA in chloroform at 60 °C for 24 h (hereafter designated as g-PBCT). PPZn fabricated using a similar methodology was investigated previously [1]. Typically, suitable contents of phenylphosphonic acid and zinc nitrate were individually dissolved in deionized water and were mixed/stirred together. Then, the additional drops of 0.1 M aqueous NaOH were added to the mixed solution to adjust the pH value of 5–6. The prepared sample was filtered, washed with water and ethanol, and dried under vacuum at 60 °C for 3 days. Chemical modification was operated by mixing organic modifier 1,6-hexanediamine and PPZn at room temperature in ethanol solution for 3 days. The prepared m-PPZn was filtered and dried at 60 °C for 3 days under vacuum.

Various contents of m-PPZn, g-PBCT and EDC as a catalyst were separately dissolved in dichloromethane, then completely mixed and mechanically stirred for 3 days. This fabricated solution was dried at 40 °C for 3 days under vacuum. The composites were classified as *x* wt% g-PBCT/m-PPZn, where *x* wt% is the weight percent of m-PPZn. These prepared g-PBCT/m-PPZn composites were hot pressed at 180 °C for 5 min. The specimens were cut into 1 × 4 cm^2^ pieces for further study.

### 2.2. Artificial Photodegradation Test

In order to investigate the effect of m-PPZn on the photodegradation behavior of g-PBCT, a UV lamp (Philips CLEO HPA 400S, Amsterdam, The Netherlands) with a radiation between 300 and 400 nm (including UV-B and UV-A) was used to irradiate the fabricated samples for 1, 2, 3, and 4 weeks. The value of irradiance evaluated at the sample surface and the dose utilized through 1-day exposure was 6 mJ/cm^2^ and 518.4 J/cm^2^, which were measured by Illuminometer and calculated for 24 h (MG 07.1, Genicom Co., Ltd., Daejeon, Republic of Korea). The relative humidity of the environment and the temperature of the sample surface were about 40% and 30 °C, respectively.

### 2.3. Analytical Procedures

Fourier transform infrared spectroscopy (FTIR) experiments were carried out in a determined range of 400–4000 cm^−1^ using a Spectrum One spectrometer (Perkin-Elmer, Waltham, MA, USA). The resolution was 4 cm^−1^ and for each sample 50 spectra were recorded. UV-Vis spectra were determined in the range of 250–400 nm by a U-3900 UV-Vis spectrophotometer (Hitachi, Tokyo, Japan). The morphology of prepared samples was identified by the field-emission scanning electron microscopy (FESEM) and transmission electron microscopy (TEM). The FESEM and TEM experiments were measured by a JEOL JSM-6700F field-emission instrument (JEOL Ltd., Tokyo, Japan) and JEOL JEM-2010 (JEOL Ltd., Tokyo, Japan), respectively. The sample of TEM experiments was prepared by a Reichert Ultracut ultramicrotome. The surface of all samples was coated by gold to avoid charging. Wide-angle X-ray diffraction (WAXD) experiments were determined using X-ray diffractometer (Bruker D8, BRUKER AXS, Inc., Madison, WI, USA) with a Ni-filtered Cu Kα radiation. The WAXD measurement was recorded 2θ ranging from 1.5° to 40° with an increment of 1°/min. Gel permeation chromatography (717 Plus HPLC Autosampler, Waters, Milford, MA, USA) with refractive index detectors (2414, Waters, Milford, MA, USA) was used to determine the molecular weights of the samples. The sample calibration used the polystyrene standards with narrow molecular-weight distributions was operated.

A UV lamp light ageing cabinet was used to measure the photodegradation of all samples for different intervals. At intervals and before the definite irradiation occurred, each film sample was analyzed by Perkin-Elmer FTIR spectrometer. The evaluation of photooxidation was by using the determination of carbonyl index (A_C=O_/A_C—H_), which was identified as the area ratio of carbonyl absorbance to the reference peak [20]. The changes in the carbonyl absorbance are characterized using the absorption peak contributed to the C=O stretching of the ester group in the region of 1600–1800 cm^−1^ and the reference peak assigned to the C—H stretching in the region of 2800–3050 cm^−1^. Therefore, the carbonyl index is calculated by the absorbance of the carbonyl peak at 1709 cm^−1^ due to carbonyl formation in the polymer to reference peak of the C—H stretching at 2856 cm^−1^.

## 3. Results and Discussion

### 3.1. Preparation and Characterization of g-PBCT/m-PPZn Composites

The FTIR spectra of PPZn, m-PPZn, and biodegradable organic modifier 1,6-hexanediamine are shown in Figure 1. For PPZn and m-PPZn, the FTIR spectra show an absorption peak at 1633 cm^−1^ and a broad band between 3400–3500 cm^−1^, which are associated to the stretching mode of monohydrates molecules in the interlayer gallery of PPZn. The absorption peaks of the phenyl C=C bands were obtained at 1700–2000 cm^−1^. The absorption peaks at 650–750 cm^−1^ and 980–1200 cm^−1^ are assigned to the phenyl ring and the PO_3_ group of phosphonic acid, respectively [13,14]. In the FTIR spectra of 1,6-hexanediamine, a broad absorption peak between 3350–3550 cm^−1^ can be attributed to the N—H stretching vibrations while the peaks of the –CH_2_– groups of 1,6-hexanediamine were observed at 2856 and 2930 cm^−1^, respectively, which was also presented in m-PPZn [21,22]. This result indicates that the surface of PPZn was successfully modified using organic modifier 1,6-hexanediamine. The UV-Vis absorbance spectra of the PPZn, m-PPZn, and organic modifier 1,6-hexanediamine are shown in Figure 2. The absorbance spectra of m-PPZn in the region of UV-A and UV-B were relatively higher than that of PPZn, indicating the better absorption ability of the m-PPZn. This result might be contributed to the higher absorption ability of organic modifier 1,6-hexanediamine in the UV-A and UV-B region.

The WAXD profiles of the m-PPZn, g-PBCT, and g-PBCT/m-PPZn composites are shown in Figure 3. According to previous investigation [12], three strong diffraction peaks at 2θ = 6.34°, 12.49°, and 18.68° were obtained for the PPZn, which characterize to the crystal planes of (010), (020), and (030) Miller index, respectively. These diffraction peaks of m-PPZn were clearly shifted to 2θ = 3.30°, 6.61°, and 8.47°, which indicates the increase in the interlayer spacings of m-PPZn. Therefore, the interlayer spacings of PPZn and m-PPZn determined via Bragg’s equation were 1.39 and 2.67 nm, respectively. These results also demonstrate that the intercalation of 1,6-hexanediamine was successfully inserted into the interlayer spacing of PPZn. For the WAXD data of g-PBCT, they present five strong diffraction peaks at 2θ = 16.10°, 17.40°, 20.66°, 23.37°, and 25.17°, which were contributed to the crystal planes of (0−11), (010), (−101), (100), and (1−11) Miller index of the polybutylene terephthalate (PBT) crystallite [5]. By the incorporation of m-PPZn content, the peak positions and intensities of the m-PPZn almost disappeared. By adding the m-PPZn content to 5 wt%, the PBCT/m-PPZn composites presented a little trace of the main diffraction peak of m-PPZn at 2θ = 3.30°. These results suggest that the crystal structure of the PBCT does not change with the addition of the m-PPZn, and the stacking structure of the m-PPZn in the g-PBCT/m-PPZn composites presents almost delaminated conformations. Moreover, Figure 4 reveals the morphology of 5 wt% g-PBCT/m-PPZn composites examined using TEM. From this image, the stacking structure of m-PPZn is not clearly noted, which is in accordance with the WAXD curves. Therefore, both WAXD and TEM data present that the g-PBCT polymer matrix was intercalated into the interlayer spacing of m-PPZn, which is approximately delaminated in the composite materials.

### 3.2. Structure, Morphology, and Physical Properties of g-PBCT/m-PPZn Composites

The UV-Vis absorbance spectra between 250–400 cm^−1^ of the PBCT and the g-PBCT/m-PPZn composite materials with various weight ratios are shown in Figure 5. The UV-Vis results of the g-PBCT material display exceptional UV-B and UV-C absorption characteristics; however, they reveal weak UV-A absorption characteristics owing to the existence of the carbonyl group [23,24]. The absorbance spectra of the g-PBCT/m-PPZn composites exhibit superior UV barrier properties than that of the g-PBCT. Moreover, the UV-A and UV-B absorption attributes are increased with increasing the m-PPZn loading. These results are contributed to the presence of excellent UV-A and UV-B absorbance properties of m-PPZn. Therefore, the absorption characteristics of the g-PBCT/m-PPZn composites increases with increasing the m-PPZn content, implying that the shielding effect of the PPZN layers as well as the incorporation of zinc element in the layers.

Figure 6 shows the FTIR spectra of g-PBCT and 5 wt% g-PBCT/m-PPZn composites at different UV irradiation times. Enlarged FTIR spectra of all samples from 1600–1850 cm^−1^ are also inserted in this figure. All FTIR spectra are very similar in both samples and the band region at 1650–1750 cm^−1^ is appointed to the C=O stretching of the ester group [20,25]. The absorption peaks C—H stretching vibrations were observed at 2856 and 2946 cm^−1^ peaks, which the peak at 2856 cm^−1^ was employed as the reference peak for the determination of the carbonyl index. During the photooxidation processes, numerous carboxyl groups and aldehyde were fabricated. Therefore, the amount of carboxyl groups altered at various photooxidation times, which is attributed to the increase and broadening of C=O absorbance peak at 1709 cm^−1^ as the irradiation time increases. Before the UV irradiation, the FTIR spectra of g-PBCT and 5 wt% g-PBCT/m-PPZn composites revealed an absorbance peak at 1709 cm^−1^ and the intensities of these peaks were slightly increased after irradiation, which implied the chemical decomposition of the g-PBCT through photodegradation process [26,27]. Similarly, the FTIR spectra of 1 wt% and 3 wt% g-PBCT/m-PPZn composites were also examined, and the results were close to that of 5 wt% g-PBCT/m-PPZn composites. All experimental data indicate that the photodegradation behaviors and mechanism of g-PBCT and various weight ratios of g-PBCT/m-PPZn composites at different UV irradiation times are similar. Figure 7 shows the FTIR spectra of g-PBCT and various weight ratios of g-PBCT/m-PPZn composites before and after 4 weeks UV irradiation times. Before the UV irradiation, the absorption peaks of carbonyl group of all samples are very close to each other. The absorption peaks of carbonyl group of the pure PBCT and various weight ratios of g-PBCT/m-PPZn composites continues to increase as the UV irradiation time increases. After 4 weeks of the UV irradiation, the comparison of the FT-IR curves for the g-PBCT/m-PPZn composites with pure g-PBCT matrix presents the increase in the C=O peak intensity for the various weight ratios of g-PBCT/m-PPZn composite is less than that of pure g-PBCT. These results reveal that the m-PPZn can significantly prevent the g-PBCT photodegradation process.

Normally, the degree of photodegradation of g-PBCT can be shown using the change in the carboxyl groups of g-PBCT [20]. The intensity ratio under the carbonyl absorbance to the reference peak appointed to the A_C=O_ and A_C—H_ group (A_C=O_/A_C—H_) was employed to calculate the development of photodegradation. A low value of A_C=O_/A_C—H_ implies less photodegradation occurrence in the g-PBCT polymer matrix. Figure 8 presents the changes in the carbonyl index of various weight ratios of g-PBCT/m-PPZn composites and pure g-PBCT at different UV irradiation times. Detail data of carbonyl index for all samples are also displayed in Table 1. It can be seen that the carbonyl index of each sample rises weekly with the irradiation process, which implies the photodegradation process occurs as the irradiation process begins. Compared to the various weight ratios of g-PBCT/m-PPZn composites, the value of carbonyl index of the pure g-PBCT is relatively higher. At the same irradiation time, the value of carbonyl index decreases gradually with increasing the content of m-PPZn in the composite. These findings reveal that the m-PPZn is a useful stabilizer for g-PBCT under irradiation and hinders the photodegradation of polymer matrix. The decrease in the carbonyl index with increasing m-PPZn component shows that the m-PPZn prevents the photodegradation of the PBCT. This enhancement in the photostability of g-PBCT with the m-PPZn indicates that the m-PPZn contains great absorbability within the UV band shown in Figure 2, which can prevent high energy UV light successfully and decrease photodegradation of the g-PBCT matrix.

Figure 9 shows the change of number-average molecular weights (Mn) of the PBCT and various weight ratios of g-PBCT/m-PPZn composites at different UV irradiation times. Details of the change of Mn for the pure PBCT and various weight ratios of g-PBCT/m-PPZn composites are also presented in Table 2. All results present a significant change of molecular weight in the first week, which decreases with increasing the m-PPZn content. The molecular weight of the pure PBCT and various weight ratios of g-PBCT/m-PPZn composites continues to decrease as the UV irradiation time increases from 1 week to 4 weeks. After 4 weeks of irradiation, the 5 wt% g-PBCT/m-PPZn retains the highest molecular weight as compared to pure g-PBCT, 1 wt% and 3 wt% g-PBCT/m-PPZn composites, exhibiting the best photodegradation resistance. In particular, the molecular weight of g-PBCT and 5 wt% g-PBCT/m-PPZn composites after photodegradation for 4 weeks was decreased by about 20.76% and 8.21% as compared to the samples before photodegradation, respectively. These results indicate that the m-PPZn can play a considerable role in photodegradation protection for PBCT polymer matrix. A similar investigation was reported by Chen et al. [28]; they found that the incorporation of inorganic layered montmorillonite (MMT) could pay a regular contribution to the UV reflection ability, causing less photon energy interacting with the biodegradable PBAT. Since a common drawback of polymers related with the outdoor purpose is the chain scission, which significantly decreases their functioning [29]. In particular, in the use of mulch films in agricultural applications, it is important to investigate the photodegradation of polymers by UV light. Therefore, this work offers an appropriate approach to enhance the g-PBCT photodegradation resistance.

To examine the effect of m-PPZn on the photodegradation behavior of g-PBCT during the photodegradation process, the SEM images were used to examine the morphological changes caused by irradiation. Figure 10 displays the SEM images of g-PBCT and various weight ratios of g-PBCT/m-PPZn composites before and after 4 weeks UV irradiation times. Before irradiation, both the g-PBCT film and g-PBCT/m-PPZn composites exhibit a relatively smooth surface. After 4 weeks of irradiation, the surface of the pure g-PBCT film found several randomly distributed cracks occurring, which is in accordance with the previous observation [25]. This surface change is owing to the degradation of the g-PBCT, probably beginning from the film surface, and then worsening in the depth dimension as irradiation time increases. According to previous data, the change of carbonyl index of the pure g-PBCT is higher than that of g-PBCT/m-PPZn composites. Therefore, the surface of 5 wt% g-PBCT/m-PPZn composite film at the same irradiation conditions changes insignificant, which indicates that the m-PPZn containing better resistance to photodegradation can efficiently inhibit the aging of the g-PBCT matrix. This result is also consistent with the change of Mn for pure g-PBCT and various weight ratios of g-PBCT/m-PPZn composites. The possible photodegradation mechanism of the g-PBCT/m-PPZn composites investigated in this study may be similar to that of biodegradable aliphatic polyester, poly(butylene succinate), which was investigated using mass spectrometry and was reported three photooxidation processes, including the hydroxyl end groups oxidation, the Norrish I of chain cleavage, and α-hydrogen abstraction [26]. Therefore, many carboxyl groups and aldehyde were generated during the processes of photodegradation, and the amount of carboxyl groups significantly altered at various irradiation times, which induces the increase and broadening of C=O peaks with the increasing irradiation time. Further study of g-PBCT/m-PPZn composites using mass spectrometry are still under investigation.

## 4. Conclusions

In this study, the application of various weight ratio of 1,6-hexanediamine modified PPZn with g-PBCT to improve UV protection was exhibited. The X-ray diffraction data suggest that the crystal structure of the PBCT does not change with the addition of the m-PPZn. Both WAXD and TEM results of the g-PBCT/m-PPZn composites showed that the m-PPZn is approximately delaminated in the composite materials. The UV barrier properties were clearly improved with the incorporation of m-PPZn content into the g-PBCT polymer matrix. All experimental measurements of the change of carbonyl index and molecular weight suggested that the photodegradation of g-PBCT decreased as m-PPZn loading increased. In particular, the molecular weight of g-PBCT and 5 wt% g-PBCT/m-PPZn composites after photodegradation for 4 weeks was decreased about 20.76% and 8.21%, respectively. This observation could be assigned to the outstanding UV reflection ability of m-PPZ. Thus, this investigation shows a typical methodology and significant advantage of fabricating the photodegradation stabilizer to enhance the UV photodegradation behavior of biodegradable polymer using a m-PPZn compared to other UV stabilizer particles or additives.

## Figures and Tables

**Figure 1 polymers-15-01276-f001:**
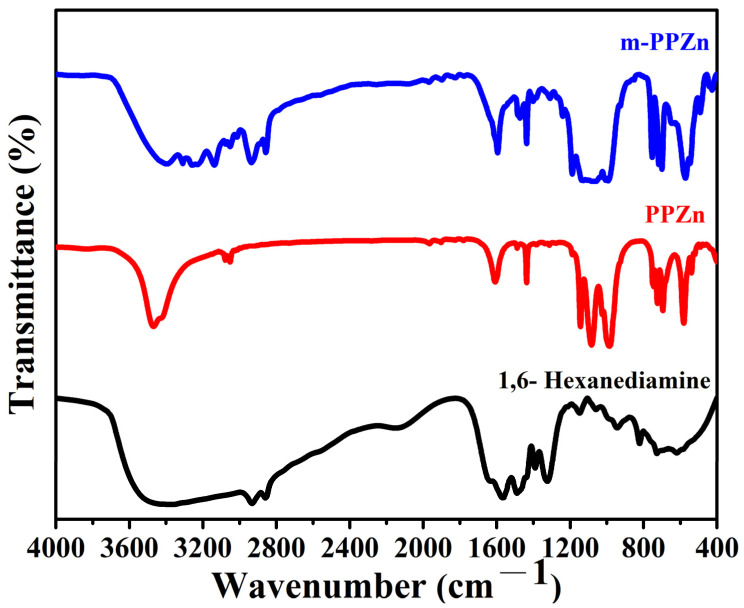
FTIR spectra of the PPZn, 1,6-hexanediamine and m-PPZn.

**Figure 2 polymers-15-01276-f002:**
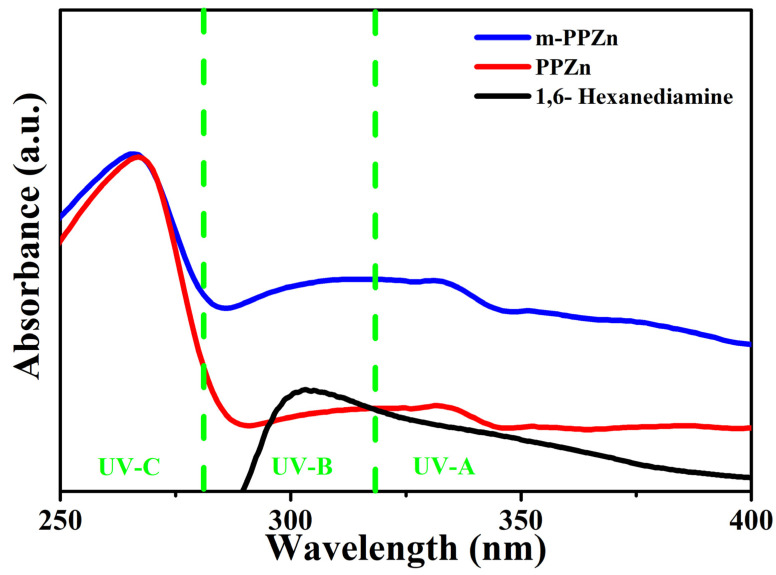
UV-Vis absorbance spectra of the PPZn, 1,6-hexanediamine and m-PPZn. The green lines in this figure illustrate the UV-A, UV-B, and UV-C regions.

**Figure 3 polymers-15-01276-f003:**
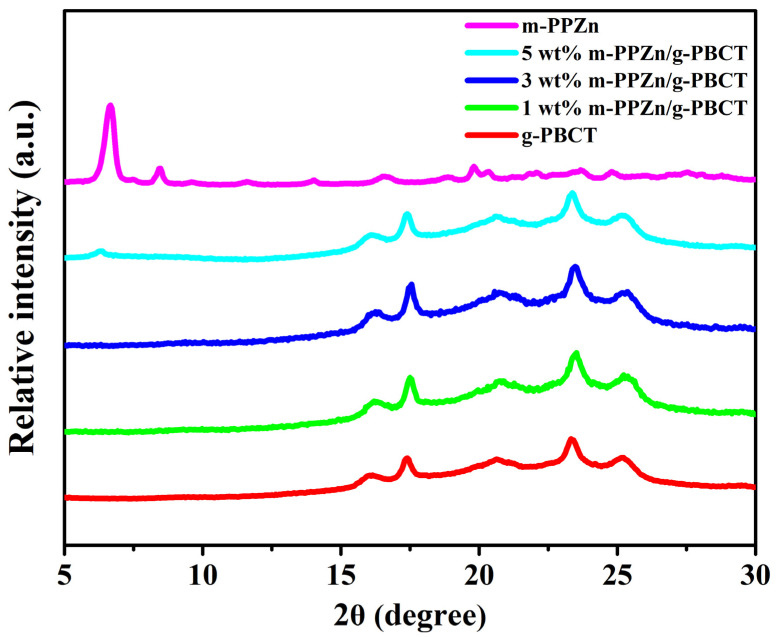
WAXD patterns of the m-PPZn, g-PBCT, and various weight ratios of g-PBCT/m-PPZn composites.

**Figure 4 polymers-15-01276-f004:**
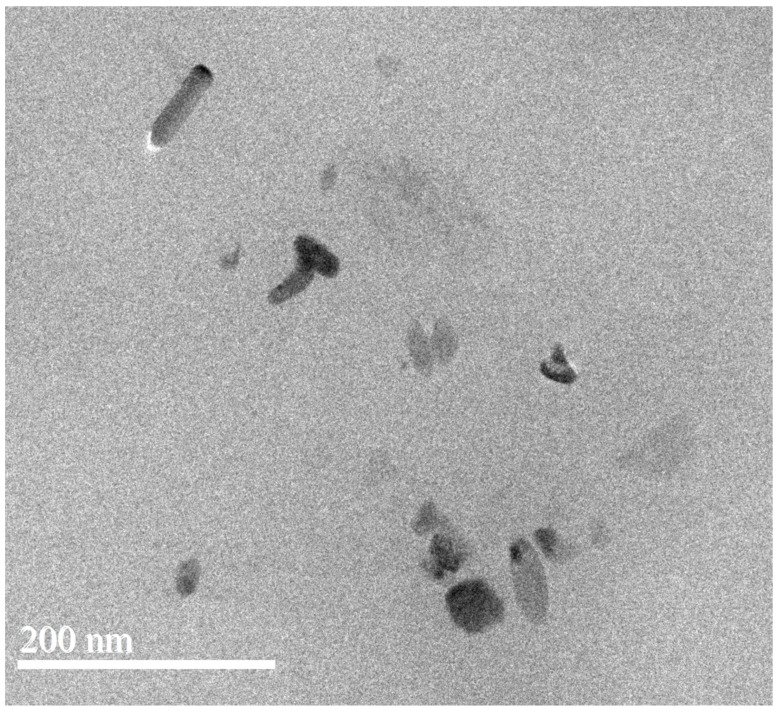
TEM images of the 5 wt% g-PBCT/m-PPZn composite.

**Figure 5 polymers-15-01276-f005:**
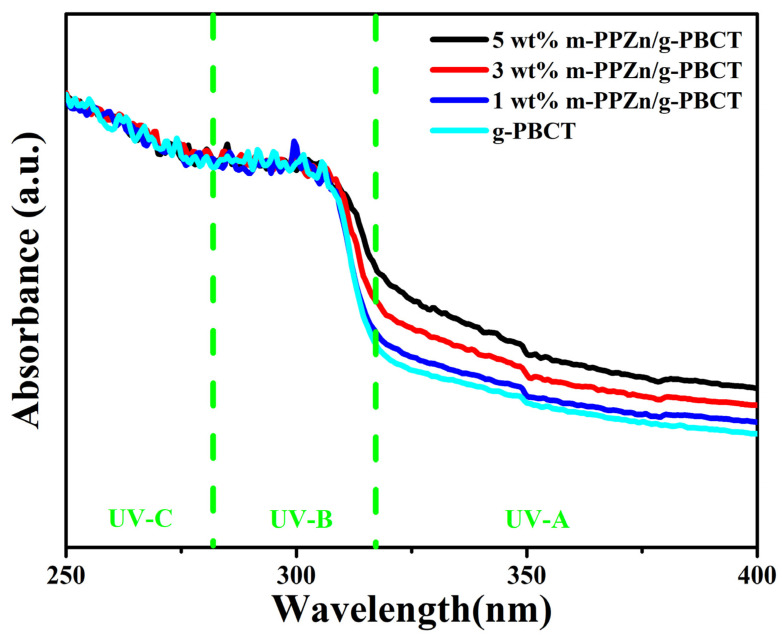
UV-Vis spectra of the g-PBCT and various weight ratios of g-PBCT/m-PPZn composites. The green lines in this figure illustrate the UV-A, UV-B, and UV-C regions.

**Figure 6 polymers-15-01276-f006:**
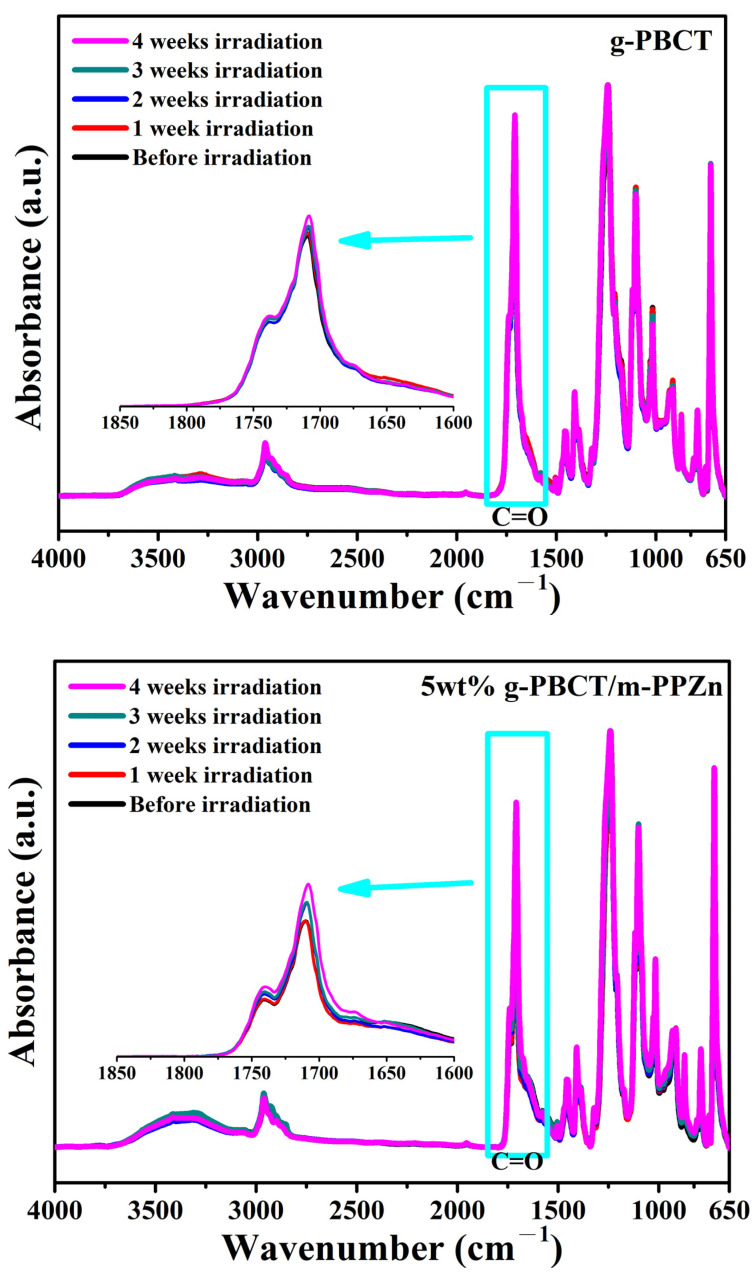
Change of FTIR spectra of g-PBCT and 5 wt% g-PBCT/m-PPZn composites nanocomposites at different UV irradiation times.

**Figure 7 polymers-15-01276-f007:**
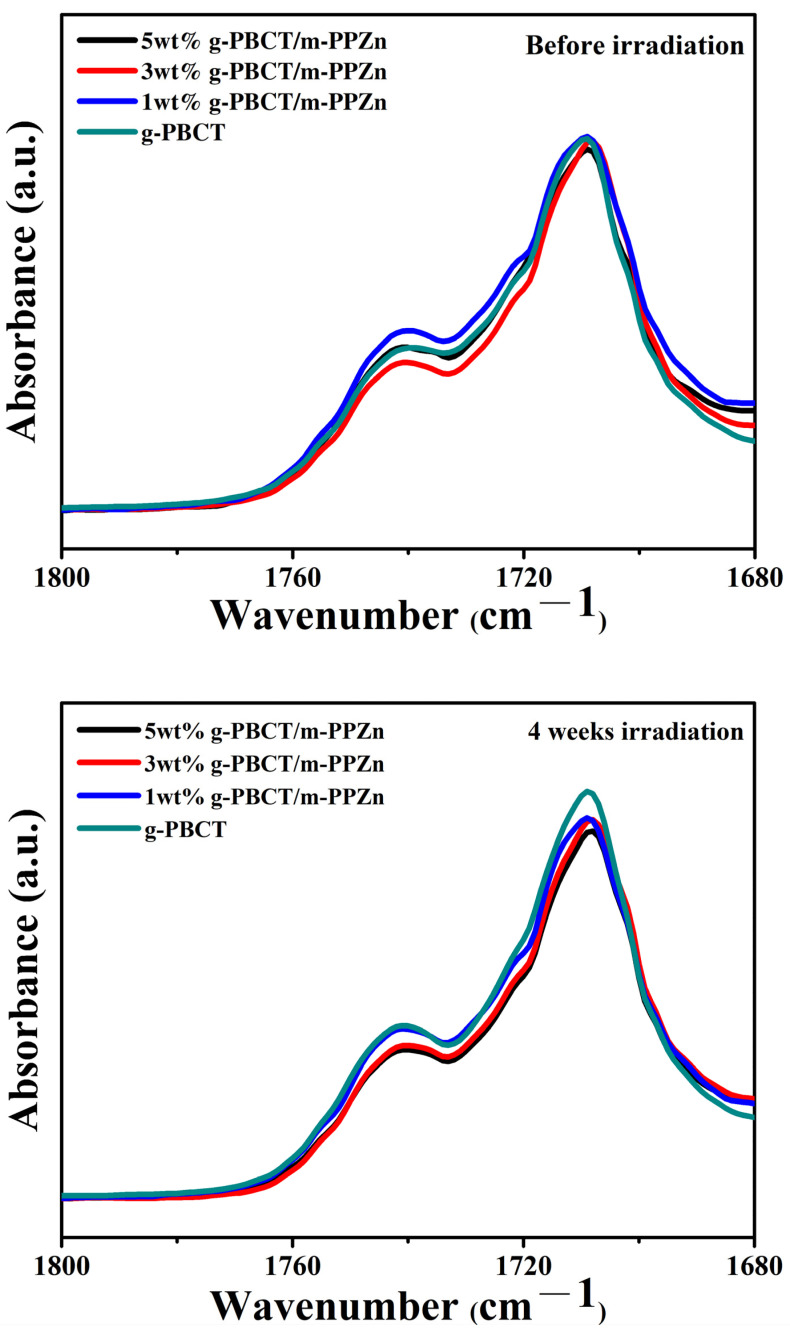
Change of FTIR spectra of g-PBCT and various weight ratios of g-PBCT/m-PPZn composites before and after 4 weeks UV irradiation times.

**Figure 8 polymers-15-01276-f008:**
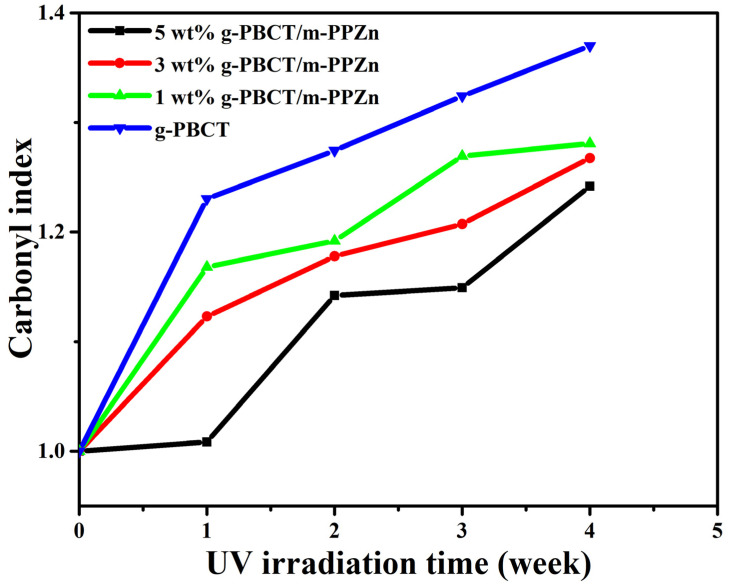
Change of carbonyl index of g-PBCT and various weight ratios of g-PBCT/m-PPZn composites at different UV irradiation times.

**Figure 9 polymers-15-01276-f009:**
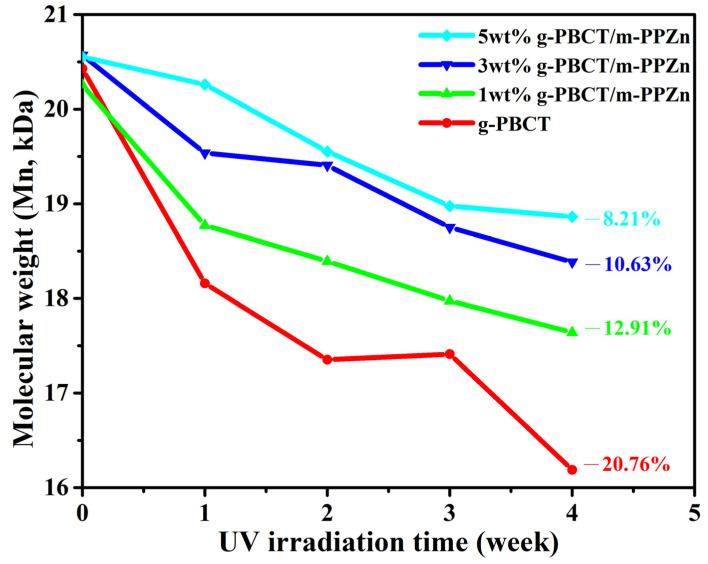
Change of number-average molecular weight (Mn) of g-PBCT and various weight ratio of g-PBCT/m-PPZn composites at different UV irradiation times.

**Figure 10 polymers-15-01276-f010:**
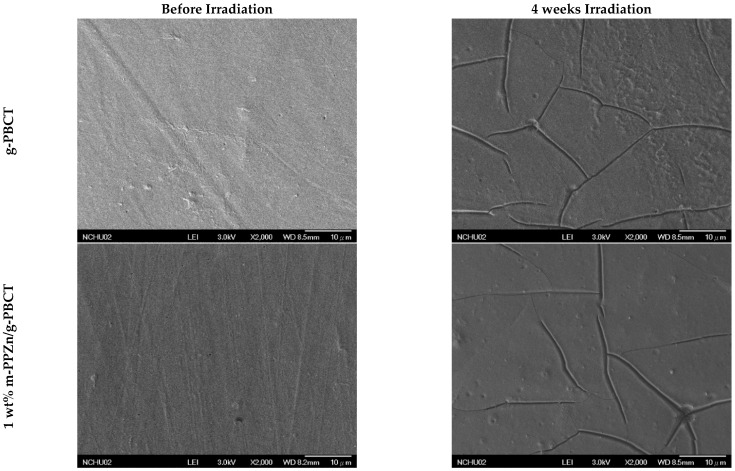
SEM images of g-PBCT and various weight ratio of g-PBCT/m-PPZn composites before and after 4 weeks UV irradiation times.

**Table 1 polymers-15-01276-t001:** Carbonyl index * of g-PBCT/m-PPZn composites after 4 weeks UV irradiation.

Sample Name	0 Week	1 Week	2 Weeks	3 Weeks	4 Weeks
g-PBCT	1	1.23	1.27	1.32	1.37
1 wt% g-PBCT/m-PPZn	1	1.17	1.19	1.27	1.28
3 wt% g-PBCT/m-PPZn	1	1.12	1.18	1.21	1.27
5 wt% g-PBCT/m-PPZn	1	1.01	1.14	1.15	1.24

* Carbonyl index = ($\frac{A_{C=O}^{t}}{A_{C—H}^{t}}$)/($\frac{A_{C=O}^{t0}}{A_{C—H}^{t0}}$); A indicates the absorbance value; t_0_ and t, respectively, before and after UV irradiation.

**Table 2 polymers-15-01276-t002:** Number-average molecular weights (Mn) of g-PBCT/m-PPZn composites after 4 weeks UV irradiation.

Sample Name	0 Week	1 Week	2 Weeks	3 Weeks	4 Weeks
g-PBCT	20.43 × 10^3^	18.16 × 10^3^	17.35 × 10^3^	17.41 × 10^3^	16.19 × 10^3^
1 wt% g-PBCT/m-PPZn	20.26 × 10^3^	18.77 × 10^3^	18.39 × 10^3^	17.97 × 10^3^	17.64 × 10^3^
3 wt% g-PBCT/m-PPZn	20.57 × 10^3^	19.54 × 10^3^	19.41 × 10^3^	18.75 × 10^3^	18.38 × 10^3^
5 wt% g-PBCT/m-PPZn	20.55 × 10^3^	20.26 × 10^3^	19.55 × 10^3^	18.98 × 10^3^	18.86 × 10^3^

## Data Availability

The data that support the findings of this study are available on request from the corresponding author.

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
