# Peer review of "Investigation on the Photodegradation Stability of Acrylic Acid-Grafted Poly(butylene carbonate-co-terephthalate)/Organically Modified Layered Zinc Phenylphosphonate Composites"

_polymers, 2023, doi:10.3390/polym15051276_

Round 1

Reviewer 1 Report (Previous Reviewer 1)

Dear,

The manuscript investigated through photodegradation of acrylic acid-grafted PBCT (g-PBCT)/organically modified layered zinc phenylphosphonate (m-PPZn). Compared to the first version, the manuscript has been improved. Minor details for the manuscript:

Page 4. “According to previous investigation, ...........”. Please add the reference to that statement;

> Authors need to present the photodegradation mechanism and the reactions of the new chemical groups, as discussed in the FTIR;

> Conclusion. "Thus, this investigation shows a typical methodology of enhancing the....". Please use the term "photodegradation stabilizer";

> Please report the advantage of this additive as a UV stabilizer, compared to the usual ones: UV absorbers, excited state deactivators, primary and secondary antioxidants;

Author Response

Response to the comments of reviewers

The authors appreciate the referees’ careful reading and thoughtful suggestions. Points by point responses to reviewer’s comments are discussed below.

Reviewer #1:

The manuscript investigated through photodegradation of acrylic acid-grafted PBCT (g-PBCT)/organically modified layered zinc phenylphosphonate (m-PPZn). Compared to the first version, the manuscript has been improved. Minor details for the manuscript:

 Answer: The authors appreciate the reviewer’s careful reading and thoughtful suggestions. Points by point responses to reviewer’s comments are discussed below.

  1. Page 4. “According to previous investigation, ...........”. Please add the reference to that statement.

Answer: The authors agree with the reviewer’s comment. The authors have added the reference on page 4, line 147 “According to previous investigation [22], three strong diffraction peaks at 2θ = 6.34º, 12.49º, and 18.68º were obtained for the PPZn, which characterize to the crystal planes of (010), (020), and (030) Miller index, respectively.” according to the reviewer’s comment.

  1. Authors need to present the photodegradation mechanism and the reactions of the new chemical groups, as discussed in the FTIR.

Answer: The authors agree with the reviewer’s comment. The authors have added the possible photodegradation mechanism as discussed in the FTIR on page 12, line 272 “The possible photodegradation mechanism of the g-PBCT/m-PPZn composites investigated in this study may be similar to that of biodegradable aliphatic polyester, poly(butylene succinate), which was investigated using mass spectrometry and was reported three photooxidation processes, including α-hydrogen abstraction, the Norrish I of chain cleavage, and the hydroxyl end groups oxidation [36].” according to the reviewer’s comment.

  1. Conclusion. "Thus, this investigation shows a typical methodology of enhancing the....". Please use the term "photodegradation stabilizer".

Answer: The authors agree with the reviewer’s comment. The authors have used the term of "photodegradation stabilizer" on page 13, line 287 “Thus, this investigation shows a typical methodology and significant advantage of fabricating the photodegradation stabilizer to enhance the UV photodegradation behavior of biodegradable polymer using a m-PPZn compared to other additives.” according to the reviewer’s comment.

  1. Please report the advantage of this additive as a UV stabilizer, compared to the usual ones: UV absorbers, excited state deactivators, primary and secondary antioxidants.

Answer: The authors agree with the reviewer’s comment. The authors have added the advantage of this additive as a UV stabilizer compared to the usual ones in the Introduction section on page 2, line 52 “In order to further inhibit the photodegradation properties of PBCT copolymer, the addition of excellent UV absorption properties of inorganic material between 250 and 400 nm, such as layered zinc phenylphosphonates (PPZn) with two-dimensional lamellar structure, acted as the protecting materials into the polymer matrix can enhance their photo barrier properties compared to the zero-dimensional UV stabilizer particles or additives.” according to the reviewer’s comment.

Reviewer 2 Report (Previous Reviewer 2)

The authors fabricated 1,6-hexanediamine modified layered zinc phenylphosphonate (m-PPZn) utilized as a UV protection additive for acrylic acid-grafted poly(butylene carbonate-co-terephthalate) (g-PBCT) were versus solution mixing process. Both wide-angle X-ray diffraction and transmission electron microscopy data show that the g-PBCT polymer matrix was intercalated into the interlayer spacing of m-PPZn. The evolution of photodegradation behavior for g-PBCT/m-PPZn composites were characterized using Fourier transform infrared spectroscopy and gel permeation chromatography after being artificially irradiated by a light source. The enhanced UV protection ability of m-PPZn in the composite materials was calculated using the change of carboxyl group produced via photodegradation. All results indicate that the carbonyl index of the g-PBCT/m-PPZn composite materials after photodegradation for 4 weeks was extensively lower than that of the pure g-PBCT polymer matrix. These results were also confirmed by the decrease in g-PBCT molecular weight as m-PPZn content increases. Both observations were probably owing to the better UV reflection ability of m-PPZn. This investigation shows a typical methodology of enhancing the biodegradable polymer UV photodegradation behavior using a m-PPZn.

The paper will be ready for publication after major revision according to the attached pdf file.

Author Response

Response to the comments of reviewers

The authors appreciate the referees’ careful reading and thoughtful suggestions. Points by point responses to reviewer’s comments are discussed below.

Reviewer #2:

The authors fabricated 1,6-hexanediamine modified layered zinc phenylphosphonate (m-PPZn) utilized as a UV protection additive for acrylic acid-grafted poly(butylene carbonate-co-terephthalate) (g-PBCT) were versus solution mixing process. Both wide-angle X-ray diffraction and transmission electron microscopy data show that the g-PBCT polymer matrix was intercalated into the interlayer spacing of m-PPZn. The evolution of photodegradation behavior for g-PBCT/m-PPZn composites were characterized using Fourier transform infrared spectroscopy and gel permeation chromatography after being artificially irradiated by a light source. The enhanced UV protection ability of m-PPZn in the composite materials was calculated using the change of carboxyl group produced via photodegradation. All results indicate that the carbonyl index of the g-PBCT/m-PPZn composite materials after photodegradation for 4 weeks was extensively lower than that of the pure g-PBCT polymer matrix. These results were also confirmed by the decrease in g-PBCT molecular weight as m-PPZn content increases. Both observations were probably owing to the better UV reflection ability of m-PPZn. This investigation shows a typical methodology of enhancing the biodegradable polymer UV photodegradation behavior using a m-PPZn.

The paper will be ready for publication after major revision according to the attached pdf file.

 Answer: The authors appreciate the reviewer’s careful reading and thoughtful suggestions. Points by point responses to reviewer’s comments are discussed below.

  1. Please highlight your contributions in introduction.

Answer: The authors agree with the reviewer’s comment. The authors have added the contributions of this research on page 2, line 63 “From our detailed research, no investigation on the photodegradation of the g-PBCT/m-PPZn nanocomposites has been reported.” according to the reviewer’s comment.

  1. Why did the authors used Fourier transform infrared spectroscopy in a 100 range of 400-4000 cm-1?

Answer: The scan range of Fourier transform infrared spectroscopy is from 400 cm-1 to 4000 cm-1 according to the experimental design of instrument. The resolution was 4 cm−1. The authors have added this information in the Analytical procedures on page 3, line 98 “Fourier transform infrared spectroscopy (FTIR) experiments were carried out determined in a range of 400-4000 cm-1 using a Spectrum One spectrometer (Perkin-Elmer, Waltham, MA, USA). The resolution was 4 cm−1 and for each sample 50 spectra were recorded.” according to the reviewer’s comment.

  1. Use Mendeley or Endnote to fix the references format.

Answer: The authors agree with the reviewer’s comment. The authors have used Endnote to fix the references format according to the reviewer’s comment.

4-1. The structure is good and the language is appropriate. 

4-2. The abstract presents a good summary of the ideas and outcomes described in the paper. 

4-3. The abstract is of reasonable length. 

4-4. The authors describe the key findings of their experiments. 

4-5. The authors have done a good job at stating the general problem of solar-powered still optimization (SPS) and their approach to dealing with this problem. 

4-6. The authors have done a good job introducing the ideas of their study and how it fits into the larger field of neural net-based process optimization. 

4-7. Overall, the introduction is good. 

4-8. The methodology section describes each of the methods in depth such that the experiments could be reproduced by another researcher. 

4-9. The details of the experimental design are well written. 

4-10. All of the sections dealing with different analytical methods were cited with historic and contemporary references. 

4-11. The results and discussion sections are well written. 

4-12. The authors present a good case for the interpretations and conclusions based on the data generated from the various analytical methods. 

4-13. The figures and tables are well-structured, concise and easy to read. 

4-14. Write concisely and precisely which changes you recommend. 

4-15. The results and discussion is well structured. 

4-16. The writing style and grammar are very good and need no improvement. 

4-17. There are no additional experiments or analyses needed for this manuscript.

Answer: The authors thanks and appreciate the reviewer’s above positive comments about this submitted manuscript.

  1. The abstract should be rewritten to reflect the significance of the proposed work. The current abstract shows a lot of background information. 

Answer: The authors agree with the reviewer’s comment. The authors have modified the abstract according to the reviewer’s comment.

  1. Conclusion: What are the advantages and disadvantages of this study compared to the existing studies in this area?

Answer: The authors agree with the reviewer’s comment. The authors have modified the Conclusion of this study on page 13, line 287 “Thus, this investigation shows a typical methodology and significant advantage of fabricating the photodegradation stabilizer to enhance the UV photodegradation behavior of biodegradable polymer using a m-PPZn compared to other additives.” according to the reviewer’s comment.

  1. Experimental design should be discussed.

Answer: The authors agree with the reviewer’s comment. The authors have detailly explain the experimental setup in the Experimental section according to the reviewer’s comment.

  1. Support the discussion and the introduction by the following papers that may help to understand the applications of polymeric materials in different fields”

Answer: The authors agree with the reviewer’s comment. The authors have added the discussion about the applications of composite materials in different fields on page 1, line 29 “Biodegradable polymers, such as poly(lactic acid),  poly(1,4-butanediol succinate), poly(butylene adipate-co-terephthalate) (PBAT), and poly(butylene carbonate-co-terephthalate) (PBCT), have recently attracted interest due to their environmentally friendly characteristics and can be used to replace traditional fossil-based polymer materials in applications such as packaging, agriculture, textile, biomedical materials, and energy harvesting composite materials.” according to the reviewer’s comment.

  1. Bistable Morphing Composites for Energy-Harvesting Applications

Answer: The authors have added 4 more references about energy harvesting composite materials in the Introduction section according to the reviewer’s comment.

10-1. An Optimized Multilayer Perceptrons Model Using Grey Wolf Optimizer to Predict Mechanical and Microstructural Properties of Friction Stir Processed Aluminum Alloy Reinforced by Nanoparticles

10-2. Predicting Characteristics of Dissimilar Laser Welded Polymeric Joints Using a Multi-Layer Perceptrons Model Coupled with Archimedes Optimizer

Answer: The authors think both comments were not directly related to this manuscript. Therefore, the authors did not modify this manuscript according to the reviewer’s both comments.

  1. The inspiration of your work must further be highlighted. Some suggested recent literatures should add. Add future works as bullets. 

Answer: The authors agree with the reviewer’s comment. The authors have modified this manuscript to show the importance of this research according to the reviewer’s comment. Some recent literatures have been added according to the reviewer’s comment. The authors have added the Future work on page 12, line 272 “The possible photodegradation mechanism of the g-PBCT/m-PPZn composites investigated in this study may be similar to that of biodegradable aliphatic polyester, poly(butylene succinate), which was investigated using mass spectrometry and was reported three photooxidation processes, including α-hydrogen abstraction, the Norrish I of chain cleavage, and the hydroxyl end groups oxidation [36]. Further study of g-PBCT/m-PPZn composites using mass spectrometry are still under investigations.” according to the reviewer’s comment.

  1. Looking and wishes for the revised version.

Answer: The authors appreciate the reviewer’s careful reading and thoughtful suggestions. The authors have modified this manuscript according to the reviewer’s comment.

Round 2

Reviewer 2 Report (Previous Reviewer 2)

Avoid lumped citation such as : “agriculture, textile, biomedical materials, and energy harvesting composite materials [1-14]”.; you may use a review paper instead of all these references.

“a Chemical Industry Co., Ltd.” Add country.

Why does the transmittance of m-PPZn is higher than others?

Why does g-PBCT polymer matrix intercalated into the interlayer spacing of m-PPZn?

Use clear TEM images, Figure 4 should be replaced.

What are the main features in SEM images in figure 10?

Author Response

Response to the comments of reviewers

The authors appreciate the referees’ careful reading and thoughtful suggestions. Points by point responses to reviewer’s comments are discussed below.

Reviewer #2:

  1. Avoid lumped citation such as : “agriculture, textile, biomedical materials, and energy harvesting composite materials [1-14]”.; you may use a review paper instead of all these references.

Answer: The authors agree with the reviewer’s comment. Four review papers have been used as references on page 1, line 35 “In contrast, for the past few years, there has been renewal of interest in biodegradable polymers, such as poly(lactic acid) (PLA),  poly(1,4-butanediol succinate) (PBS), poly(butylene adipate-co-terephthalate) (PBAT), poly(butylene succinate-co-terephthalate) (PBST), and poly(butylene carbonate-co-terephthalate) (PBCT), due to their environmentally friendly characteristics and can be used to replace traditional fossil-based polymer materials in applications such as packaging, agriculture, textile, biomedical materials, and energy harvesting composite materials [1-4].” according to the reviewer’s comment.

  1. “a Chemical Industry Co., Ltd.” Add country.

Answer: The authors have added the country of all chemical companies in the Materials on page 2, line 81 “Phenylphosphonic acid (C6H5P(O)(OH)2, >98%), sodium hydroxide (NaOH, >98%), zinc nitrate hexahydrate (Zn(NO3)2·6H2O, >99%), 1-ethyl-3-(3-dimethylaminopropyl)carbodiimide (C8H17N3·HCl, EDC, >98%), and 1,6-hexanediamine (H2N(CH2)6NH2, >97%) were obtained from Sigma-Aldrich (St. Louis, MO, USA). Acrylic acid (C3H4O2, AA, >99%) was acquired from Tedia Company, Inc. (Fairfield, MO, USA). 2,2’-Azobis-isobutyronitrile (C8H12N4, AIBN, >98%), chloroform (CHCL3, >99%) was purchased from Showa Chemical Industry Co., Ltd. (Minato-Ku, Tokyo, Japan). 1,4-butanediol (C4H10O2, BD, >99%), dimethylene carbonate (C3H6O3, DMC, >98%), and dimethylene terephthalate (C10H10O4, DMT, >99%) were obtained from Alfa Aesar Chemical Company (Ward Hill, MA, USA).” according to the reviewer’s comment.

  1. Why does the transmittance of m-PPZn is higher than others?

Answer: Because PPZn contains two-dimensional lamellar structure can act as acted as the photo protecting materials, the absorbance spectra of PPZn and m-PPZn in the region of UV-C were close to each other. But the absorbance spectra of m-PPZn in the region of UV-A and UV-B were relatively higher than that of PPZn, which might be contributed to the higher absorption ability of organic modifier 1,6-hexanediamine in this region.

  1. Why does g-PBCT polymer matrix intercalated into the interlayer spacing of m-PPZn?

Answer: Because the g-PBCT contains carboxylic acid (COOH) functional group on the polymer backbone of PBCT, which can produce the chemical reaction with amine (NH2) group of m-PPZn. The authors have explained this mechanism on page 2, line 58 “Among these reactions, the acrylic acid containing carboxyl functional group is commonly used in such grafting process. In order to further inhibit the photodegradation properties of PBCT copolymer, the addition of excellent UV absorption properties of inorganic material between 250 and 400 nm, such as layered zinc phenylphosphonates (PPZn) with two-dimensional lamellar structure, acted as the protecting materials into the polymer matrix can enhance their photo barrier properties compared to the zero-dimensional UV stabilizer particles or additives [10,11]. Since PPZn contains a coordinated water molecule in the interlayered spacing, the loss of this water molecule provides possible intercalation or catalytic reactions with an open coordination site on the zinc atom [12,13]. Thus, organo-modifiers are necessary to enlarge the interlayered spacing of PPZn and to generate better compatibility between the polymer matrix and PPZn [14].” according to the reviewer’s comment.

  1. Use clear TEM images, Figure 4 should be replaced.

Answer: The authors agree with the reviewer’s comment. The clear TEM image has been replaced according to the reviewer’s comment.

  1. What are the main features in SEM images in figure 10?

Answer: Because the photodegradation of g-PBCT/m-PPZn composites will induce the change of molecular weight, the SEM images were used to examine the morphological changes during this process. The authors have explained this feature on page 12, line 299 “To investigate the effect of m-PPZn on the photodegradation behavior of g-PBCT during the photodegradation process, the SEM images were used to examine the morphological changes caused by irradiation.” according to the reviewer’s comment.

Round 3

Reviewer 2 Report (Previous Reviewer 2)

Accept

Author Response

Response to the comments of reviewers

The authors appreciate the referees’ careful reading and thoughtful suggestions. Points by point responses to reviewer’s comments are discussed below.

Reviewer #2:

  1. Accept

Answer: The authors thanks and appreciate the referees’ careful reading and thoughtful suggestions.

This manuscript is a resubmission of an earlier submission. The following is a list of the peer review reports and author responses from that submission.

Round 1

Reviewer 1 Report

Dear,

The manuscript presented an investigation through the phodegradation of PBCT grafted with acrylic acid (g-PBCT)/organically modified layered zinc phenylphosphonate (m-PPZn). This is a good investigation for the literature database. Some corrections must be met before publication:

> Introduction. Authors need to indicate the novelty of the manuscript. Furthermore, a specific review should be added, indicating recent advances in this area;

> Authors need to update the references as well as increase the quantity;

> Materials. Please inform the degree of purity of each reagent, as well as the concentration;

> Page 2. Artificial Photodegradation Test. Please inform the type of lamp: UVA or UVB?
> “The value of irradiance evaluated at the level of the sample surface and the dose used during 1-day exposure was 6 mJ/cm2 and 518.4 J/cm2”. Inform how the radiation control was carried out;
> “Fourier transform infrared spectroscopy (FTIR) experiments in a range...”. Please enter scan quantity and resolution;
> Please improve the quality of Figure 4, as it was cropped in the manuscript;
> Authors need to present, discuss and propose the degradation mechanism of the developed compounds. The photodegradation reactions must be presented;
> The authors must correlate the surface obtained by SEM with chemical modification due to photodegradation;
Author Response

Response to the comments of reviewers

The authors appreciate the referees’ careful reading and thoughtful suggestions. Points by point responses to reviewer’s comments are discussed below.

Reviewer #1:

The manuscript presented an investigation through the photodegradation of PBCT grafted with acrylic acid (g-PBCT)/organically modified layered zinc phenylphosphonate (m-PPZn). This is a good investigation for the literature database. Some corrections must be met before publication:

 Answer: The authors agree with the reviewer’s comment. The authors have modified this manuscript according to the reviewer’s comment.

  1. Introduction. Authors need to indicate the novelty of the manuscript. Furthermore, a specific review should be added, indicating recent advances in this area.

Answer: The authors agree with the reviewer’s comment. The authors have added the novelty of this research on page 2, line 60 “From our detailed research, no study on the photodegradation of the g-PBCT/m-PPZn nanocomposites has been reported, thus, this is novel research.” according to the reviewer’s comment. The authors have added one review paper in this research area according to the reviewer’s comment. The information of this review paper is also listed below.

[Ref] Progress in Polymer Science, 2020, 109:101291. “Crystallization of biodegradable and biobased polyesters: Polymorphism, cocrystallization, and structure-property relationship.”

  1. Authors need to update the references as well as increase the quantity.

Answer: The authors agree with the reviewer’s comment. More references published recently have added in the Introduction section according to the reviewer’s comment.

  1. Materials. Please inform the degree of purity of each reagent, as well as the concentration.

Answer: The authors agree with the reviewer’s comment. The authors have added the degree of purity of each reagent in the Experimental section according to the reviewer’s comment.

  1. Page 2. Artificial Photodegradation Test. Please inform the type of lamp: UVA or UVB?

Answer: The authors agree with the reviewer’s comment. The authors have added the type of lamp in the Experimental section on page 2, line 91 “In order to investigate the effect of m-PPZn on the photodegradation behavior of g-PBCT, a UV lamp (Philips CLEO HPA 400S, Amsterdam, The Netherlands) with a radiation between 300 and 400 nm (including UV-B and UV-A) was used to irradiate the fabricated samples for 1, 2, 3, and 4 weeks.” according to the reviewer’s comment.

  1. “The value of irradiance evaluated at the level of the sample surface and the dose used during 1-day exposure was 6 mJ/cm2 and 518.4 J/cm2”. Inform how the radiation control was carried out.

Answer: The authors agree with the reviewer’s comment. The value of irradiance evaluated at the level of the sample surface was directly measured by Illuminometer for 30 mins and then calculated for 24 h. The authors have added this information in the Experimental section on page 2, line 94 “The value of irradiance evaluated at the level of the sample surface and the dose used during 1-day exposure was 6 mJ/cm2 and 518.4 J/cm2, which were measured by Illuminometer and calculated for 24 h (MG 07.1, Genicom Co., Ltd., Daejeon, Korea).” according to the reviewer’s comment.

  1. “Fourier transform infrared spectroscopy (FTIR) experiments in a range...”. Please enter scan quantity and resolution.

Answer: The authors agree with the reviewer’s comment. The authors have added the scan quantity and resolution on page 3, line 101 “The resolution was 4 cm−1 and for each sample 50 spectra were recorded.” according to the reviewer’s comment.

  1. Please improve the quality of Figure 4, as it was cropped in the manuscript.

Answer: The authors agree with the reviewer’s comment. The authors have improved the quality of Figure 4 according to the reviewer’s comment. The TEM image is also shown below.

  1. Authors need to present, discuss and propose the degradation mechanism of the developed compounds. The photodegradation reactions must be presented.

Answer: The authors agree with the reviewer’s comment. The authors have proposed possible degradation mechanism of fabricated materials on page 12, line 271 “The possible photodegradation mechanism of the g-PBCT/m-PPZn composites may be similar to that of biodegradable aliphatic polyester, poly(butylene succinate), which was investigated using mass spectrometry and was reported three photooxidation processes, including α-hydrogen abstraction, the Norrish I of chain cleavage, and the hydroxyl end groups oxidation [36]. Further study of g-PBCT/m-PPZn composites using mass spectrometry are still under investigations.” according to the reviewer’s comment. Detail degradation mechanism needs further experiments to analyze the chemical compounds of fabricated materials using mass spectrometry after irradiation.

  1. The authors must correlate the surface obtained by SEM with chemical modification due to photodegradation.

Answer: The authors agree with the reviewer’s comment. The authors have added the discussion of the SEM images versus photodegradation on page 12, line 266 “According to previous data, the change of carbonyl index of the pure g-PBCT is higher than that of g-PBCT/m-PPZn composites. Therefore, , the surface of 5 wt% g-PBCT/m-PPZn composite film at the same irradiation conditions changes insignificant, which indicates that the m-PPZn containing better resistance to photodegradation can efficiently inhibit the aging of the g-PBCT matrix.” according to the reviewer’s comment.

Reviewer 2 Report

All results indicate that the carbonyl index of the g-PBCT/m-PPZn composite materials after photodegradation for 4 weeks was extensively lower than that of the pure g-PBCT polymer matrix. These results were also confirmed by the decrease in g-PBCT molecular weight as m-PPZn content increases. Both observations were probably owing to the better UV reflection ability of m-PPZn. This investigation shows a typical methodology of enhancing the biodegradable polymer UV photodegradation behavior using a m-PPZn.

The paper may be published  after major modifications.

“Biodegradable polymers, such as poly(lactic acid), poly(1,4-butanediol succinate), poly(butylene adipate-co-terephthalate) (PBAT), and poly(butylene carbonate-co-terephthalate) (PBCT) have attracted interest in recent year [1-4].”, why did they attract the interest of the reserchers?

“owing to the absence of acidic compounds during its in vivo degradation and photodegradation”, what else. Mention all advantages and disadvantages.

“. In this study, we focus on the UV protection effect of m-PPZn on g-PBCT. To stimulate the photodegradation of the g-PBCT/m-PPZn samples, an artificial light source was provided. The carbonyl index, number-average molecular weight, and morphology of prepared samples were used to estimate the UV protection of m-PPZn in the g-PBCT polymer matrix.”, the novelty of the study should be deeply discussed compared with other studies.

“The decrease in the carbonyl index with increasing m-PPZn content demonstrates that the m-PPZn retards the photodegradation of the PBCT” , how?

Quality of figures are excellent.

Explain the experimental setup in details.

Future work must be included.

 “To study the effect of surface changes on the photodegradation process, the SEM measurements were used to examine the morphological changes caused by irradiation. Figure 10 displays the SEM images of g-PBCT and various weight ratios of g-PBCT/mPPZn composites before and after 4 weeks UV irradiation times. Before irradiation, both the g-PBCT film and g-PBCT/m-PPZn composites exhibit a relatively smooth surface.”, revise this  paragraph.

The introduction should be supported by recent publications to show the importance of composite materials in different engineering applications such as in energy harvesting especially from MDPI:

Bistable Morphing Composites for Energy-Harvesting Applications

Future work must be included.

Looking and wishes for the revised version.

 Author Response

Response to the comments of reviewers

The authors appreciate the referees’ careful reading and thoughtful suggestions. Points by point responses to reviewer’s comments are discussed below.

Reviewer #2:

All results indicate that the carbonyl index of the g-PBCT/m-PPZn composite materials after photodegradation for 4 weeks was extensively lower than that of the pure g-PBCT polymer matrix. These results were also confirmed by the decrease in g-PBCT molecular weight as m-PPZn content increases. Both observations were probably owing to the better UV reflection ability of m-PPZn. This investigation shows a typical methodology of enhancing the biodegradable polymer UV photodegradation behavior using a m-PPZn.

The paper may be published  after major modifications.

 Answer: The authors agree with the reviewer’s comment. The authors have modified this manuscript according to the reviewer’s comment.

  1. “Biodegradable polymers, such as poly(lactic acid), poly(1,4-butanediol succinate), poly(butylene adipate-co-terephthalate) (PBAT), and poly(butylene carbonate-co-terephthalate) (PBCT) have attracted interest in recent year [1-4].”, why did they attract the interest of the researchers?

Answer: The authors agree with the reviewer’s comment. The authors have added the attracted reason in the Introduction section on page 1, line 29 “Biodegradable polymers, such as poly(lactic acid),  poly(1,4-butanediol succinate), poly(butylene adipate-co-terephthalate) (PBAT), and poly(butylene carbonate-co-terephthalate) (PBCT), have recently attracted interest due to their environmentally friendly characteristics and can be used to replace traditional fossil-based polymer materials in applications such as packaging, agriculture, textile, biomedical materials, and energy harvesting composite materials” according to the reviewer’s comment.

  1. “owing to the absence of acidic compounds during its in vivo degradation and photodegradation”, what else. Mention all advantages and disadvantages.

Answer: The authors agree with the reviewer’s comment. The authors have added the advantages and disadvantages in the Introduction section on page 1, line 36 “Compared with aliphatic-aromatic PBAT, PBCT contains more promising comprehensive properties and competitive cost with low melting temperature and crystallization rate and is more feasible candidates for biomedical and package applications owing to the absence of acidic compounds during its in vivo degradation and photodegradation.” according to the reviewer’s comment.

  1. “. In this study, we focus on the UV protection effect of m-PPZn on g-PBCT. To stimulate the photodegradation of the g-PBCT/m-PPZn samples, an artificial light source was provided. The carbonyl index, number-average molecular weight, and morphology of prepared samples were used to estimate the UV protection of m-PPZn in the g-PBCT polymer matrix.”, the novelty of the study should be deeply discussed compared with other studies.

Answer: The authors agree with the reviewer’s comment. The authors have added the novelty of this research on page 2, line 64 “From our detailed research, no study on the photodegradation of the g-PBCT/m-PPZn nanocomposites has been reported, thus, this is novel research.” according to the reviewer’s comment.

  1. “The decrease in the carbonyl index with increasing m-PPZn content demonstrates that the m-PPZn retards the photodegradation of the PBCT” , how?

Answer: The authors have discussed the role of the carbonyl index in the photodegradation on page 9, line 210 “Normally, the carboxyl groups can be used as a signature to reveal the degree of photodegradation of g-PBCT [21]. The intensity ratio under the carbonyl absorbance to the reference peak appointed to the AC=O and AC-H group (AC=O/AC-H) was employed to calculate the development of photodegradation. A high value of AC=O/AC-H implies more photodegradation occurrence in the g-PBCT polymer matrix.” according to the reviewer’s comment.

  1. Quality of figures are excellent. Explain the experimental setup in details. Future work must be included.

Answer: The authors agree with the reviewer’s comment. The authors have detailly explain the experimental setup in the Experimental section according to the reviewer’s comment. The authors have added the future work of this study on page 12, line 271 “The possible photodegradation mechanism of the g-PBCT/m-PPZn composites may be similar to that of biodegradable aliphatic polyester, poly(butylene succinate), which was investigated using mass spectrometry and was reported three photooxidation processes, including α-hydrogen abstraction, the Norrish I of chain cleavage, and the hydroxyl end groups oxidation [36]. Further study of g-PBCT/m-PPZn composites using mass spectrometry are still under investigations.” according to the reviewer’s comment.

  1. “To study the effect of surface changes on the photodegradation process, the SEM measurements were used to examine the morphological changes caused by irradiation. Figure 10 displays the SEM images of g-PBCT and various weight ratios of g-PBCT/mPPZn composites before and after 4 weeks UV irradiation times. Before irradiation, both the g-PBCT film and g-PBCT/m-PPZn composites exhibit a relatively smooth surface.”, revise this  paragraph.

Answer: The authors agree with the reviewer’s comment. The authors have revised this paragraph on page 10, line 258 “To investigate the effect of m-PPZn on the photodegradation behavior of g-PBCT during the photodegradation process, the SEM images were used to examine the morphological changes caused by irradiation. Figure 10 displays the SEM images of g-PBCT and various weight ratios of g-PBCT/m-PPZn composites before and after 4 weeks UV irradiation times. Before irradiation, both the g-PBCT film and g-PBCT/m-PPZn composites exhibit a relatively smooth surface.” according to the reviewer’s comment.

  1. The introduction should be supported by recent publications to show the importance of composite materials in different engineering applications such as in energy harvesting especially from MDPI:

Answer: The authors agree with the reviewer’s comment. The authors have added the discussion of composite materials in different engineering applications such as in energy harvesting on page 1, line 29 “Biodegradable polymers, such as poly(lactic acid),  poly(1,4-butanediol succinate), poly(butylene adipate-co-terephthalate) (PBAT), and poly(butylene carbonate-co-terephthalate) (PBCT), have recently attracted interest due to their environmentally friendly characteristics and can be used to replace traditional fossil-based polymer materials in applications such as packaging, agriculture, textile, biomedical materials, and energy harvesting composite materials.” according to the reviewer’s comment. Four more references from MDPI have added in this manuscript and are also listed below.

  1. Covaci, C.; Gontean, A. Piezoelectric energy harvesting solutions: A review. Sensors 2020, 20, 3512.
  2. Stiubianu, G.-T.; Bele, A.; Bargan, A.; Potolinca, V.O.; Asandulesa, M.; Tugui, C.; Tiron, V.; Hamciuc, C.; Dascalu, M.; Cazacu M. All-polymer piezo-composites for scalable energy harvesting and sensing devices. Molecules 2022, 27, 8524.
  3. Trigona, C.; Graziani, S.; Pasquale, G.D.; Pollicino, A.; Nisi, R.; Licciulli, A. Green energy harvester from vibrations based on bacterial cellulose. Sensors 2020, 20, 136.
  4. Dong, X.; Liu, Z.; Yang, P.; Chen, X. Harvesting wind energy based on triboelectric nanogenerators. Nanoenergy Adv. 2022, 2, 245–268.

  1. Bistable Morphing Composites for Energy-Harvesting Applications

Answer: The authors have added 4 more references about energy harvesting composite materials from MDPI according to the reviewer’s comment.

  1. Future work must be included.

Answer: The authors agree with the reviewer’s comment. The authors have added the Future work on page 12, line 271 “The possible photodegradation mechanism of the g-PBCT/m-PPZn composites may be similar to that of biodegradable aliphatic polyester, poly(butylene succinate), which was investigated using mass spectrometry and was reported three photooxidation processes, including α-hydrogen abstraction, the Norrish I of chain cleavage, and the hydroxyl end groups oxidation [36]. Further study of g-PBCT/m-PPZn composites using mass spectrometry are still under investigations.” according to the reviewer’s comment.

  1. Looking and wishes for the revised version.

Answer: The authors appreciate the reviewer’s careful reading and thoughtful suggestions. The authors have modified this manuscript according to the reviewer’s comment.
